# Protocol for a Prospective Cohort Study on Determinants of Outcomes in Lumbar Radiculopathy Surgery

**DOI:** 10.3390/healthcare13192444

**Published:** 2025-09-26

**Authors:** Alejandro Aceituno-Rodríguez, Carlos Bustamante, Carmen Rodríguez-Rivera, Miguel Molina-Álvarez, Carlos Rodríguez-Moro, Rafael García-Cañas, Carlos Goicoechea, Luis Matesanz-García

**Affiliations:** 1Faculty of Health Sciences, Escuela Internacional de Doctorado, Universidad Rey Juan Carlos, 28922 Alcorcón, Spain; alejandro.aceituno@universidadeuropea.es; 2Department of Physiotherapy, Faculty of Medicine, Health and Sports, European University of Madrid, 28670 Madrid, Spain; 3Servicio de Análisis de Clínicos, Hospital Clínico San Carlos, 28040 Madrid, Spain; carlosbustamante.lopez@salud.madrid.org; 4Area of Pharmacology, Nutrition and Bromatology, Department of Basic Health Sciences, Unidad Asociada I+D+i Instituto de Química Médica (IQM) CSIC-URJC, Universidad Rey Juan Carlos, 28922 Alcorcón, Spain; carmen.rodriguez@urjc.es (C.R.-R.); carlos.goicoechea@urjc.es (C.G.); 5High Performance Experimental Pharmacology Research Group (PHARMAKOM), Faculty of Health Sciences, Universidad Rey Juan Carlos, 28922 Alcorcón, Spain; 6CranioSPain Research Group, Centro Superior de Estudios Universitarios La Salle, Universidad Autónoma de Madrid, 28023 Madrid, Spain; miguel.molina@lasallecampus.es; 7Cognitive Neuroscience, Pain, and Rehabilitation Research Group (NECODOR), Faculty of Health Sciences, Universidad Rey Juan Carlos, 28922 Alcorcón, Spain; 8Servicio de Cirugía Ortopédica y Traumatología, Hospital Central de la Defensa “Gómez Ulla”, 28047 Madrid, Spain; crodmo1@oc.mde.es (C.R.-M.); rgarc18@oc.mde.es (R.G.-C.)

**Keywords:** sciatic nerve, radiculopathy, neurofilament proteins, treatment outcome, MicroRNAs

## Abstract

**Introduction**: Lumbar radiculopathies involving the entrapment of nerve roots in the lumbar spine are common neuropathic conditions. These conditions affect 40% to 70% of individuals in their lifetime and lead to significant medical costs. **Objective:** This study aims to identify clinical, psychological, and biomarker-based prognostic factors that predict functional outcomes following surgery for lumbar radiculopathy. **Materials and Methods**: This prospective cohort study, conducted at Hospital Central de la Defensa Gómez Ulla, Madrid (Spain), adheres to the STROBE guidelines. The study includes patients aged 18–75 with lumbar radiculopathy, confirmed by clinical diagnosis, imaging, and electromyography (EMG) findings. Exclusion criteria include previous lumbar spine surgeries and systemic diseases. The primary outcome is the Oswestry Low Back Pain Disability Questionnaire. Sample size calculations, based on a conservative effect size (f^2^ = 0.20), determined the need for 172 participants, accounting for a 15% dropout rate and 80% power. **Procedure:** Patients will undergo an initial assessment, including EMG tests, sociodemographic and psychological questionnaires, blood sample tests, and physical questionnaires. This process will be repeated six months post-intervention, except for the blood sample test, expectations questionnaire, and EMG, which will be performed only once. **Statistical Analyses:** Data will be analyzed using Python 3.12.3, employing a multivariate linear regression analysis. Assumptions of linearity, independence, homoscedasticity, normality, and no multicollinearity will be validated. Corrective measures will be applied if assumptions are violated. **Ethics and Dissemination**: The study follows the Declaration of Helsinki guidelines and has been approved by the Ethics Committee of Universidad Rey Juan Carlos (070220241052024). Potential risks will be minimized, and adverse events will be recorded and addressed. Findings will be published in high-impact journals and presented at conferences.

## 1. Introduction

Lumbar radiculopathies involving the entrapment of nerve roots in the lumbar spine stand as one of the most prevalent neuropathic conditions worldwide. In the UK, around 40% to 70% of individuals will suffer from this condition at some point in their lives; hence, these ailments account for significant medical costs, estimated at £270 million [1] and reaching $305 million in the USA [2].

The NeuPSIG working group has recently redefined radiculopathy as “a lesion or disease of a nerve root or dorsal root ganglia associated with a condition slowing or block” [3], marking a pivotal shift in its understanding. Lumbar radicular pain, as a primary symptom, typically manifests as back pain radiating to the lower limbs [4]. The term “sciatica” is defined as a common condition related to lumbar radicular pain, but to date, no precise definition has been found for its diagnosis, giving rise to varying prevalence rates [5,6].

The diagnosis of these conditions primarily implies neurological evaluations and magnetic resonance imaging (MRI), the latter being the most evidence-supported tool for confirmation [7,8]. However, inconclusive MRI results require supplementary tests like electrodiagnostic assessments [9]. Furthermore, employing dermatomal somatosensory evoked potentials (DSEPs) offers insights into sensory responses at various spinal levels [10]. In research contexts, quantitative sensory testing (QST) serves as a psychophysical method to quantify sensory experiences [11].

Treatment modalities for lumbar radiculopathies are broadly categorized into conservative and invasive approaches. Conservative treatments, mainly involving physiotherapy and pharmacotherapy, are preferred, especially in cases with neuropathic pain [12]. Medications like pregabalin, antidepressants, and gabapentin are effective in the short to medium term but pose challenges for long-term use due to potential side effects [13]. Physiotherapy techniques, including mechanical traction [14,15], neural mobilization [16], stabilization exercises, electrotherapy, and manipulation [17] yield short- to medium-term benefits in pain relief, strength, and mobility, although their long-term efficacy remains inconsistently proven [18]. A recent meta-analysis, however, points to the high heterogeneity and potential bias in physiotherapy trials [19].

In more severe or persistent cases that are unresponsive to conservative treatments, invasive procedures like neurosurgical decompression followed by an instrumented posterolateral arthrodesis are used. Post-surgery, patients often report a rapid alleviation in pain and disability in the short term, but mild to moderate symptoms can persist for up to five years [20]. A significant concern is the incidence of “chronic pain after spinal surgery” (CPSS), ranging from 10% to 40%, characterized by recurrent pain post-surgery [21]. However, surgery is indicated in the presence of severe or progressive neurological deficits or persistent symptoms that do not respond to conservative treatment [22]. Schmid et al. have observed a lack of knowledge and values to be taken into account when selecting patients for successful surgery [6].

Biomarkers play a crucial role in the diagnosis and clinical management of peripheral nerve-related disorders, which have traditionally relied on neurophysiological parameters such as nerve conduction studies and electromyography (EMG) [23]. They are essential for assessing the severity and prognosis of peripheral neuropathies [24].

Among the different biomarkers, neurofilament light chain (NfL) is the most extensively studied. It serves as a prognostic indicator in both plasma and cerebrospinal fluid (CSF), highlighting its versatility across biological fluids [25]. NfL is a structural element of neurons in both the central and peripheral nervous systems, and axonal damage leads to its release into the interstitial fluid and subsequently into the peripheral blood [26]. Elevated NfL levels reflect axonal degeneration, which may contribute to the development of neurodegenerative disorders [27]. Increased NfL has been consistently reported in neurodegenerative [25], inflammatory [23], peripheral nerve diseases, and chemotherapy-induced neuropathies [28]. However, its role in traumatic peripheral nerve injuries remains poorly understood.

MicroRNAs also represent promising biomarkers. In particular, elevated plasma miR-30c-5p levels have been associated with the onset of neuropathic pain in chronic peripheral ischemia [29], suggesting their potential prognostic value in entrapment neuropathies.

Clusterin has been proposed as another molecular marker of disease progression and pain mechanisms. Its levels are significantly reduced in patients with painful conditions such as degenerative scoliosis [30], carpal tunnel syndrome [31], chronic widespread pain [32], and osteoarthritis [33]. Importantly, clusterin appears to return to normal after effective treatment [30], providing an objective measure of pain modulation and a potential therapeutic target.

In summary, despite evidence of NfL and microRNAs in various peripheral nerve diseases, a significant gap remains in understanding their prognostic role in traumatic peripheral nerve injuries and entrapment neuropathies.

To address this gap, we will evaluate prognostic factors related to NfL, miR-30c-5p, and clusterin in a prospective cohort of patients with lumbar radiculopathy. By integrating biomarker profiles with clinical indicators, pain measures, and patient beliefs, this study aims to refine the prediction of surgical outcomes and inform personalized treatment strategies.

The novelty of this study lies in its multidimensional design: to our knowledge, no previous protocol has simultaneously examined NfL, clusterin, and miR-30c-5p alongside psychosocial measures in patients undergoing surgery. This integrative approach seeks to advance predictive modeling in neuropathic pain and surgical outcomes.

## 2. Materials and Methods

The following methodology is presented in accordance with the STROBE guidelines [34]. A prospective cohort study will be conducted at the Hospital Central de la Defensa Gómez Ulla. Standardized procedures will be applied. All patients will receive informed consent forms prior to enrollment.

The cohort will track patients who meet the inclusion and exclusion criteria from their initial consultation through a follow-up period. For the recruitment of the sample, data will be collected from patients who meet the inclusion and exclusion criteria.

For sample selection, the following inclusion criteria will be considered: (1) age between 18–75 years old; (2) symptoms duration of more than 3 months; (3) clinical diagnosis of lumbar radiculopathy: conduction block along a spinal nerve or nerve root, clinically manifested by sensory loss in a dermatome or myotatic weakness or reflex changes [35]; (4) leg pain in L5 or S1 dermatomes; (5) clinically relevant demonstrable abnormality in imaging studies indicating compression of L5 or S1 nerve roots at the L4–L5 or L5-S1 levels; (6) positive EMG findings; and (7) listed for surgery.

The exclusion criteria will be as follows: (1) sensorimotor deficits of the femoral and/or femoral cutaneous nerve; (2) age greater than 75 or less than 18 years old; (3) previous lumbar spine surgeries; (4) cauda equina syndrome; (5) pregnancy; (6) musculoskeletal comorbidities such as rheumatoid arthritis and/or fibromyalgia; (7) systemic diseases such as diabetes mellitus and/or thyroid diseases; (8) recent chemotherapy treatment; (9) vascular diseases; (10) numbness and/or tingling in the feet preceded or accompanied by sensory alterations in the hands (polyneuropathies); (11) previous diagnosis of chronic or neuropathic pain; and (12) difficulty understanding the Spanish language.

### 2.1. Surgical Procedure Description

The surgical procedure employed in this study consists of neurosurgical decompression followed by instrumented posterolateral arthrodesis using pedicle screw fixation. Decompression is first performed to relieve pressure on the affected nerve roots. The procedure then progresses to posterolateral arthrodesis, which aims to achieve spinal stability by creating a bony fusion between adjacent vertebrae. Instrumentation is achieved with pedicle screws, strategically placed to secure the vertebrae. These screws provide immediate stabilization and facilitate the fusion process by maintaining proper alignment and immobilization of the spinal segments involved. This technique has shown substantial efficacy in the treatment of a wide range of spinal pathologies, including degenerative conditions and trauma-related instabilities [36,37].

### 2.2. Sample Description

#### 2.2.1. Sociodemographic Information

Sociodemographic data for the cohort will include age, sex, gender, ethnicity, educational level, employment status, marital status, and income level. Age and sex will be recorded separately. Gender and ethnicity will be categorized based on self-identification. Educational level will be documented according to the highest degree obtained, ranging from no formal education to advanced degrees (e.g., high school diploma, bachelor’s degree, master’s degree, doctorate). Employment status will include categories such as employed, unemployed, retired, and student. Marital status will be classified as single, married, divorced, or widowed. Income level will be reported in predefined brackets to capture the range of socioeconomic statuses within the cohort.

#### 2.2.2. Psychological Variables

Fear-avoidance and catastrophizing: The Spanish version of the Pain Catastrophizing Scale (PCS) will be used to assess the level of catastrophizing. The PCS consists of 13 items, and each item is scored on a scale from 0 to 4. The total score range is from 0 to 52, where higher scores indicate higher levels of catastrophizing. This instrument has demonstrated the same original factorial structure, comprising three factors (rumination, magnification, and helplessness), as well as adequate psychometric properties [38].

To evaluate patients’ beliefs about the impact of physical activity and work on their pain, the Fear-Avoidance Beliefs Questionnaire (FABQ) will be administered. The FABQ consists of 16 items, each rated on a seven-point Likert scale (0 = totally disagree, 6 = totally agree) [39,40]. The Spanish version has demonstrated excellent test–retest reliability (ICC = 0.97) [39]. Two subscales are scored—physical activity and work—by summing the corresponding items, with higher scores indicating stronger avoidance beliefs and a greater risk of pain-related disability.

To assess fear of movement or (re)injury related to pain, the Tampa Scale of Kinesiophobia (TSK) validated in Spanish will be used. The 11-item version of the TSK, which demonstrated good reliability properties in patients with pain, will be employed. Each item is rated on a four-point Likert scale ranging from “strongly agree” to “strongly disagree.” Total scores range from 11 to 44, with higher scores indicating more fear of movement and/or (re)injury. The minimal detectable change for chronic pain in the English version is 5.6 points [41].

### 2.3. Main Outcome

Oswestry Low Back Pain Disability Questionnaire is the gold standard for assessing functional disability in low back pain and has been validated in Spanish. This questionnaire consists of 10 questions with 5 possible responses each [42]. The results are reported as a percentage, with higher percentages indicating greater functional limitation. The questionnaire has a sensitivity of 76% and specificity of 63% for a cutoff point of ≥10 points. Additionally, it has a Cronbach’s alpha coefficient of 0.85 and an intraclass correlation coefficient for test–retest reliability of 0.92 [43].

### 2.4. Potential Predictors

#### 2.4.1. NfL

NfL concentration in serum samples will be analyzed using a SIMOA immunoassay (Quanterix, Billerica, MA, USA). The sample concentrations will be calculated based on individual standard curves for each analysis using Simoa HD-X software (version 3.1.2011.30002, Quanterix). The coefficient of variation between analyses for the two levels of quality control will be 6.4% and 11.9% [44].

#### 2.4.2. Clusterin

Serum clusterin levels shall be analyzed by immunoassay. Plasma samples shall be thawed and diluted 1:1000 to quantify clusterin concentration by ELISA (EHCLU kit, Thermo-Fischer Scientific, Frederick, MD, USA) following the manufacturer’s instructions [45].

#### 2.4.3. miR-30c-5p

For miR-30c-5p measurement, RNA extraction utilized the RNeasy Mini Kit (Qiagen, Venlo, The Netherlands), adhering to the manufacturer’s guidelines. RNA purity and concentration were assessed using a NanoDrop 2000 spectrophotometer (Thermo Fisher Scientific, Waltham, MA, USA), with a 260/280 nm absorbance ratio confirming integrity. cDNA synthesis was conducted from 5 ng of total RNA for each DRG side, using PrimeScript^TM^ RT Master Mix (Perfect Real Time) (Takara Bio Inc., Shiga, Japan), according to the manufacturer’s instructions. qPCR amplification was performed with TB Green^®^ Premix Ex Taq^TM^ (Tli RNaseH Plus) (Takara Bio Inc., Shiga, Japan). The differentially expressed miRNAs we will be looking for are as follows [29].

#### 2.4.4. Electromyography (EMG)

The EMG procedure will involve the placement of a disposable needle electrode in the affected muscles, depending on the compromised nerve root and its corresponding myotomes, to assess their bioelectrical activity. Measurements will be carried out on sensory nerve conduction velocity (m/s), distal motor latency (ms), sensory nerve action potential (SNAP) (mV), and compound motor action potential (CMAP) (mV) based on the protocol by Paige C. Roy et al. [46].

#### 2.4.5. Neurological Exam

The patellar and Achilles reflexes will be evaluated to observe possible reduction or attenuation [47].

A muscular balance assessment will be conducted, evaluating the strength of dorsiflexors, plantar flexors, knee flexors, and knee extensors using the Daniels scale [48].

#### 2.4.6. Neuropathic Pain

The PainDETECT questionnaire will be used for the identification and detection of neuropathic pain compared to other types of pain, with particular attention to the presence of back pain. This questionnaire consists of 4 sections and has been validated in Spanish. It includes the Visual Analog Scale (VAS) and a body diagram to help localize and quantify pain. The PainDETECT has a Cronbach’s alpha coefficient of 0.86 and an intraclass correlation coefficient for test–retest reliability of 0.93. For a cutoff point of ≥17 points, it has a sensitivity of 75% and specificity of 84% [49].

#### 2.4.7. Pressure Pain Threshold (PPT)

PPT is defined as the minimum amount of pressure required to cause pain. Algometry has been described as highly reliable for measuring pressure pain threshold (ICC = 0.91, 95% CI 0.82–0.97) [50]. PPT will be performed using a digital algometer (Model FDX 10^®^, Wagner Instruments, Greenwich, CT, USA). This instrument measures pressure in kilograms, with the pain threshold expressed in kg/cm^2^. In the protocol, the patient will indicate when he/she first experiences the onset of pain, at which time the pressure will be stopped and recorded. Three measurements shall be taken, with an interval of 30 s between each measurement to avoid a temporal summation effect. The measurement shall be taken at a point located at the L5 dermatome bilaterally, as well as at the region of greatest pain indicated by the patient. The mean of the three measurements shall be calculated.

#### 2.4.8. Temporal Summation

Induced temporal summation is the increase in C-fiber evoked responses by neurons in the dorsal horn of the spinal cord due to activation of C-fibers by repeated high-frequency stimuli [51]. The measurement of temporal summation is mechanical. The Pinprick stimulator set (MRC Systems GmbH, Heidelberg, Germany) is used. Pain perception is compared using the VAS for two types of stimuli: a single sharp stimulus and a train of 10 sharp stimuli with the same force (256 Nm), applied to the midpoint of the L5 dermatome and the area of greatest pain reported by the patient at a repetition frequency of 1 Hz. Both stimuli are applied over an area of 1 cm^2^. The stimuli are alternated five times within the predefined area. The ratio is calculated by dividing the average pain produced by the stimuli in train by the pain produced by the single stimulus. This method has been used and validated in several studies on temporal summation. In this study, the measurement will be performed before and after treatment [52].

#### 2.4.9. Mechanical Detection Thresholds

Mechanical detection thresholds will be obtained using von Frey filaments. They shall be applied perpendicular to the skin at a point in the L5 dermatome region for 1 s, and each filament shall be applied three times in ascending order. The smallest filament that elicits a pressure sensation shall be considered the detection threshold and shall be used to measure the patient’s sensitivity, pain, and allodynia [53].

#### 2.4.10. Vibration Detection Threshold

The vibration threshold shall be obtained using a tuning fork with a vibration frequency of 64 Hz and a scale of 8/8. To verify the threshold values, the struck and vibrating tuning fork is placed in the test area, if possible, over a bony eminence. The test subject indicates when they stop feeling the vibration of the tuning fork. The intensity of the stimulus is then plotted on the tuning fork scale. After a threefold determination of the vibration detection threshold, the arithmetic mean value of the thresholds can be calculated [54].

#### 2.4.11. Expectations

Patients will be given a pre-intervention survey requesting expectations for treatment. In this survey, the patient will be given 5 possible answers to discuss their expectations for the success of their treatment: “definitely yes”, “yes”, “not sure”, “no”, and “definitely no”. The questions that will be asked are related to the following factors [55]:Complete relief of symptoms (pain, stiffness, numbness, weakness, stability);Moderate relief of symptoms (pain, stiffness, stiffness, numbness, weakness, and stability);Performing more domestic activities;Sleeping more comfortably;Returning to usual work;Exercise and more recreational activities;Preventing future disabilities.

### 2.5. Procedure

After the primary selection of subjects by physicians, patients will be informed about the study characteristics. During this meeting, the voluntary nature of their participation will be explained, and they will be informed that they can withdraw from the cohort at any time. They will also be informed that refusing to participate will not negatively impact their treatment or scheduled follow-up with their physician.

The researcher in charge will register any dropouts or treatment failures. To prevent dropouts, one of the researchers will make phone calls to reassess and remind patients of their upcoming appointments. On the assessment day, patients will undergo all the tests and measurements described in the subsequent section on variables. The procedure will be as follows:

Within 2 weeks prior to the visit, the patient will undergo an EMG test. During the visit, patients will complete all the sociodemographic, psychological, and predictor questionnaires. After filling out the forms, the participant will undergo the following tests: mechanical detection thresholds, pressure pain threshold assessment, conditioned pain modulation, and temporal summation assessment.

At the end of the tests, the participants will go to another room where a nurse from the center will take a blood sample that will be analyzed by a professional in the laboratory. A peripheral venous blood sample (10 mL) will be collected in an EDTA tube. Thirty minutes after collection, plasma will be separated by centrifugation at 1000 g for 10 min at 4 °C. The plasma will be transferred to a 2 mL Eppendorf and stored at −80 °C until processed.

This process of predictive tests will be repeated six months after the intervention, except for the blood sample, expectations, and EMG, which will only be performed at the first session.

To minimize participant burden, all assessments will be clustered into one comprehensive preoperative visit and one follow-up visit at six months. Additional strategies include telephone reminders, flexible scheduling, and logistical support (e.g., transportation assistance if required). Adherence and dropout rates will be actively monitored, and study logistics will be adapted if necessary to maintain data completeness and participant engagement.

### 2.6. Sample Size

Sample size was calculated using Python version 3.12.3 [56]. The primary outcome of this study is the Oswestry Low Back Pain Disability Questionnaire, modeled with nine predictor variables: five measured once and four measured at two time points. The significance level was set at 0.05, power at 0.80, and the expected effect size at f^2^ = 0.20. Using a two-tailed test, the corresponding critical values were obtained. For predictors measured at multiple time points, an intraclass correlation coefficient (ICC) of 0.5 was assumed, yielding a design effect of 1.5. The initial calculation, based on the multiple linear regression formula and the total number of predictors, indicated a required sample size of 97. After adjustment for the design effect, the sample size increased to 146. Finally, accounting for an anticipated dropout rate of 15%, the recruitment target was set at 172 participants.

The choice of an expected effect size of f^2^ = 0.20 was based on three considerations. First, clinical meaningfulness: a 10-point change in the Oswestry Low Back Pain Disability Questionnaire is widely regarded as the minimal clinically important difference, corresponding to a moderate standardized effect. Second, empirical plausibility: an f^2^ of 0.20 corresponds to ~17% explained variance (R^2^ ≈ 0.17), which is consistent with prognostic models in lumbar surgery that typically report R^2^ values between 0.15 and 0.25. Third, methodological robustness: powering for this effect size explicitly accounts for potential attenuation from biomarker and questionnaire measurement error and helps ensure stable coefficient estimates with nine predictors. Taken together, these considerations render f^2^ = 0.20 both clinically relevant and statistically conservative for the planned analysis.

### 2.7. Statistical Analyses

The data will be analyzed using Python version 3.12.3 [56]. Before fitting the regression models, key assumptions will be validated: linearity between the dependent and independent variables, independence of observations, homoscedasticity, normality of residuals, and no perfect multicollinearity among predictors. Diagnostic tests and plots will be used to check these assumptions: scatter plots for linearity, the Durbin–Watson test for autocorrelation, residual plots and the Breusch–Pagan test for homoscedasticity, and the Shapiro–Wilk test and Q-Q plots for normality. Variance inflation factors (VIFs) will assess multicollinearity.

If assumptions are violated, corrective actions will be taken by adding polynomial or interaction terms for non-linearity, using time series models or generalized least squares for autocorrelation, applying robust standard errors or transforming the dependent variable for heteroscedasticity, transforming data for non-normal residuals, and addressing multicollinearity by removing or combining predictors or using regularization techniques like ridge regression or lasso.

Results will be reported as regression coefficients with 95% confidence intervals and *p*-values, with significance set at *p* < 0.05. The adjusted R-squared value will assess model fit. Sensitivity analyses will examine the robustness of findings, including excluding outliers and using alternative ICC values. All analyses will be interpreted considering clinical and statistical significance.

## 3. Discussion

Predictive models have become essential tools for assessing and anticipating the success of treatments across various areas of medicine. Based on clinical data—such as sensory, physical, demographic, psychological information, and biomarkers—these models make it possible to identify patterns and trends that help predict a patient’s response to specific therapeutic interventions. However, many current models present limitations that hinder their clinical applicability, as highlighted by the systematic review by Wynants et al. [57]. One such limitation is particularly evident in the field of neuropathic pain, where the lack of an objective measurement method complicates diagnosis and the selection of effective treatments. This underscores the need to develop objective and quantifiable tools, among which blood-derived biomarkers emerge as promising candidates. These biomarkers—genetic, neurophysiological, or molecular in nature—could not only facilitate diagnosis but also assess the effectiveness of therapies and predict individual treatment responses [58,59]. Their integration into predictive models would pave the way for more personalized pain medicine, reducing trial and error, lowering costs, and enhancing patient experience [60].

Among the potential biomarkers is NfL, which, despite limited evidence due to the lack of representative reference values to account for physiological increases—values that may also vary with patient age [61]—appears to be a promising blood biomarker for evaluating nerve damage in patients with peripheral polyneuropathy [62,63]. Elevated plasma levels of NfL have been observed in comparison to healthy individuals, suggesting its potential as a marker of neuronal injury. This overexpression has been widely documented in metabolic, inflammatory, or hereditary diseases such as diabetic polyneuropathy and Charcot–Marie–Tooth disease [64]. However, its application in traumatic injuries and entrapment neuropathies remains an emerging field.

In recent years, interest has grown in using these biomarkers as prognostic factors in various pathologies [65]. The influence of NfL has been studied in the progression of diseases related to the central nervous system [66,67], and even in infectious pathologies [68], though there is less evidence regarding its role in traumatic conditions. One area that requires further research is its utility as a predictor of neuropathy severity and postoperative progression. In this regard, some studies have shown a positive correlation between plasma NfL levels and the degree of nerve damage in patients with lumbar radiculopathies [69]. This may suggest that the biomarker could play a relevant role in evaluating treatment response. Nevertheless, additional studies are needed to determine its applicability in clinical practice, particularly in monitoring patients who undergo nerve decompression surgery or peripheral nerve repair [70].

This study aims to address this knowledge gap by evaluating the utility of NfL as a predictor of therapeutic success following surgery and, consequently, its potential use as an indicator of responders and non-responders to surgical interventions. If a positive association is found, further research will be needed to determine its diagnostic accuracy, temporal patterns, and prognostic value in other conditions.

In this way, the incorporation of biomarkers such as NfL into clinical practice could complement traditional clinical criteria, offering an objective assessment of nerve involvement and improving the identification of patients who would truly benefit from an intervention. This tool would not only enable earlier and more precise detection of neurodegenerative processes but also allow for more dynamic monitoring of neuronal damage progression over time. Furthermore, its use could optimize therapeutic decision-making, enabling more personalized, biologically informed medicine, ultimately contributing to better clinical outcomes and more efficient healthcare resource management.

## 4. Limitations

Several limitations of this study should be considered. Recruiting the target sample size of 172 patients may prove challenging within a single-center setting. The study also involves a comprehensive assessment battery, including biomarkers, EMG, questionnaires, QST, PPT, and temporal summation, which may place a considerable burden on participants and increase the risk of dropout. To mitigate this, assessments will be carefully clustered, and participants will be provided with logistical support throughout the study.

Another limitation lies in the follow-up period, which is currently limited to six months. While sufficient for short-term outcomes, this duration may not capture longer-term changes that are clinically relevant in the management of radiculopathy. Future studies should consider extending follow-up to 12–24 months to better evaluate sustained effects.

Regarding biomarker selection, NfL is supported by robust evidence, whereas clusterin and miR-30c-5p have more limited validation in radiculopathy. This could potentially reduce the external validity of the findings, and any conflicting evidence should be acknowledged. Finally, certain assessments, such as EMG, algometry, and QST, are operator-dependent. The absence of formal calibration between raters may affect reproducibility and reliability, highlighting the need for careful standardization in future research.

## 5. Ethics and Dissemination

### 5.1. Ethics

The study will be conducted in accordance with ethical procedures for medical research involving human subjects, following the guidelines of the Declaration of Helsinki, as adopted by the 18th World Medical Assembly [71] and its subsequent revisions, including the revised version from the 63rd Assembly [71]. The study has been approved by the Ethics Committee of Universidad Rey Juan Carlos (070220241052024).

### 5.2. Risks and Safety Measures

Potential risks associated with the study will be minimized through stringent safety measures. A dedicated record of all adverse effects reported by patients during treatment sessions or re-evaluations will be maintained. Participants will be monitored closely, and any adverse events will be promptly addressed according to established medical protocols. The safety and well-being of participants will be a top priority throughout the study.

### 5.3. Communication of Findings

Upon completion of the study, a comprehensive statistical analysis of the results will be performed. The findings will be prepared for publication to contribute to the body of knowledge in the field of pain and rehabilitation. Additionally, selected results will be presented at national and international conferences, ensuring wide dissemination among the scientific community and healthcare professionals. The communication strategy will also include providing feedback to study participants and collaborating institutions, reinforcing the study’s commitment to transparency and knowledge sharing.

## 6. Conclusions

This protocol outlines a prospective, multidimensional cohort study designed to evaluate biomarker, psychological, and clinical predictors of outcomes after lumbar radiculopathy surgery. Particular emphasis is placed on plasma NfL, a sensitive marker of axonal injury, given its potential to provide unique insights into the extent of neural damage and the trajectory of recovery after surgical decompression. By systematically integrating NfL with additional biomarkers (miR-30c-5p, clusterin), psychosocial scales, and quantitative sensory testing, this study aims to develop a robust prognostic model. The expected results will refine patient stratification, improve the prediction of functional recovery, and ultimately advance precision medicine in the management of neuropathic pain and lumbar spine surgery.

Given the multifactorial nature of the phenomena under study, a multidisciplinary approach is essential. Accurate interpretation of the findings requires not only methodological rigor but also the integration of clinical and laboratory perspectives. Therefore, results should be interpreted in close collaboration with surgeons, physical therapists, nurses, and laboratory scientists. This multidisciplinary framework will ensure that outcomes are translated into more precise and effective clinical interventions, thereby enhancing both the scientific relevance and the practical applicability of the study.

## Data Availability

No new data were created or analyzed in this study. Data sharing is not applicable to this article.

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
