# Peer review of "Protocol for a Prospective Cohort Study on Determinants of Outcomes in Lumbar Radiculopathy Surgery"

_healthcare, 2025, doi:10.3390/healthcare13192444_

Round 1
Reviewer 1 Report
Comments and Suggestions for Authors
Abstract
- Does not clearly highlight the main hypothesis or the initial scientific justification.
- The expected sample size or statistical power are not made explicit, which is essential in a protocol.
- Review keywords that are mesh terms. Review abbreviations
Introduction
- It points out the current knowledge gaps: lack of consensus on criteria and little long-term evidence. However, the bibliographic review is limited: few key studies are cited and not sufficiently contextualized with recent systematic reviews or international guides.
- It would be useful to highlight more the originality of the study: what does this protocol contribute compared to other cohorts already published in lumbar radiculopathy?
Methods
- Calculation of sample size and statistical power: it is not clearly described.
- Surgical heterogeneity: it is not clear how the variability in surgical techniques between hospitals/surgeons will be controlled.
- Although they mention regressions, there is a lack of detail on how they will handle relevant clinical and socioeconomic variables.
- Standardization of follow-up: although there are scheduled visits, it is not clear if all centers will apply the same collection protocol.
Discussion and Ethics and difusion
- The discussion is clear, but it is recommended to explore the limitations in greater depth and mention the need for interdisciplinary collaboration (rehabilitation, physical therapy, occupational health).
- Neither the management of conflicts of interest nor the possible role of external funding in the dissemination of results is mentioned.
- It is not explained whether any dissemination mechanism will be used to non-specialized health professionals (e.g. clinical guides, documents for primary care doctors or physiotherapists).
Author Response
Dear Reviewer 1,
We sincerely appreciate your valuable comments and suggestions, which have significantly contributed to improving the clarity, methodological rigor, and clinical applicability of our manuscript. Below, we provide our responses, clearly separating each reviewer’s comment from our reply, while maintaining a fluent narrative style.
Comment 1: The main hypothesis or initial scientific rationale is not clearly highlighted.
Response: Thank you for your comment. We have restructured the biomarker section to first present the most relevant markers (NfL, miR-30c-5p, clusterin), followed by their clinical rationale. In addition, we reformulated the main objective into shorter and more direct sentences. Lines 90-127
Comment 2: The expected sample size or statistical power is not explicitly stated.
Response: Thank you for the observation. We have included in the abstract the sample size calculation (172 participants). Line 33-35
Comment 3: Review keywords to ensure they are MeSH terms.
Response: Thank you for the comment. The keywords have been reviewed and replaced with appropriate MeSH terms. Lines 47-48
Comment 4: The literature review is limited.
Response: Thank you for the suggestion. We have expanded the introduction, citing recent systematic reviews and international guidelines (NICE 2020; Schmid 2023; Coquelet 2025, among others) to better contextualize existing knowledge gaps. Line 50-127
Comment 5: It would be useful to better emphasize the originality of the study.
Response: Thank you for the suggestion. We now highlight that this protocol is original in integrating molecular biomarkers, neurophysiological measures, and psychosocial factors into a predictive model of surgical outcomes-something not previously conducted in lumbar radiculopathy cohorts. Line 123-127
Comment 6: Sample size calculation and statistical power are not clearly described.
Response: We appreciate the reviewer’s observation. In the original submission we reported an effect size of f² = 0.35. Upon further consideration, and to adopt a more conservative and empirically grounded assumption, we have revised the calculation to use f² = 0.20 (R² ≈ 0.17). The manuscript has been updated accordingly. Lines 331-351
Comment 7: Surgical heterogeneity across hospitals/surgeons.
Response: Thank you for the comment. We have added that participating the hospital (Hospital Gómez Ulla) will follow a standardized surgical protocol (neurodecompression more instrumented fusion), and that potential technical variations will be documented for subsequent analysis. Also, we have clarified that recruitment will take place at Hospital Central de la Defensa Gómez Ulla. Line 153-163
Comment 8: Few clarifications on the handling of clinical and socioeconomic variables.
Response: Thank you for the suggestion. We now specify that sociodemographic and clinical variables (age, sex, educational level, employment status…) will be included as covariates in the multivariate regression models. Line 166-174
Comment 9: Standardization of follow-up is not clear.
Response: We thank the reviewer for this important observation. To minimize participant burden, all assessments will be clustered into a single comprehensive preoperative visit and one follow-up visit at 6 months. In addition, we will implement supportive strategies including telephone reminders, flexible scheduling, and assistance with transportation when needed. The research team will actively monitor adherence and dropout rates throughout the study, allowing us to adjust logistics promptly if unanticipated difficulties arise. We believe these measures will substantially reduce participant burden while ensuring the integrity and completeness of the data collected. Line 325-328
Comment 10: Explore limitations in more depth.
Response: Thank you for your comment. We have added an explicit subsection titled “Limitations” within the Discussion, detailing: potential bias due to the high number of assessments, risk of dropout during follow-up, the 6-month follow-up period, and the challenge of reaching the target sample size within a single center. Line 422-441
Comment 11: Mention interdisciplinary collaboration.
Response: Thank you for the observation. We now specify that results should be interpreted and applied in collaboration with traumatologists, physical therapists, nurses, and laboratory staff. Line 477-483
Comment 12: No mention of conflict of interest management or external funding.
Response: Thank you for the observation. We have clarified that no conflicts of interest exist and that, although no external funding was received, resources will be sought to support open-access dissemination of results. Line 495
Comment 13: No explanation of dissemination for non-specialist professionals.
Response: We thank the reviewer for highlighting this point. In Section 5.3 we describe our dissemination strategy, which includes peer-reviewed publications and presentations at national and international conferences to reach both the scientific community and healthcare professionals. We have now clarified that dissemination will also target non-specialist professionals—such as general practitioners, physiotherapists, and nurses—through presentations in multidisciplinary meetings, summaries in professional association newsletters, and accessible materials provided to collaborating institutions. In addition, study participants will receive feedback in lay terms. These steps reinforce our commitment to transparency and knowledge transfer beyond the specialist community. Lines 457-464
All reviewer comments have been incorporated, thereby enhancing the clarity, rigor, and clinical applicability of the manuscript. We once again thank the reviewers for their detailed and constructive feedback.
Sincerely,
Reviewer 2 Report
Comments and Suggestions for Authors
The study is quite valuable in terms of its contributions to the literature. While the multidimensional data collection approach is a strength, the methodology section is quite detailed, and some sections may require simplification and clarification. In particular, a rationale for the sample size calculation, a clearer emphasis on limitations, and support for the statistical analysis plan with references would make the study more robust.
Comments and Suggestions for Authors:
This study protocol is a comprehensive and updated design to examine clinical, psychological, and biomarker-based factors that predict outcomes after lumbar radiculopathy surgery. Considerations include:
Abstract
- The abstract contains relevant details about the study, but the aim of the study is not included. A brief aim should be included.
Introduction
- This is a reasonably detailed literature review. The section providing information on biomarkers is overly disorganized. The aim of the study could be emphasized with more concise statements.
Material and methods
The use of the STROBE Guide in the study is a strength.
- Information on how the total FABQ score was obtained should be included. Furthermore, interpreting a high or low score should emphasize its meaning (high score = more avoidance, etc.).
- The sample size calculation was provided, but the source of the effect size was not clearly explained. It should be stated why 0.35 was chosen.
- There are too many measurement tools. This can increase participant burden and increase the risk of dropouts. Has a strategy been established to address this?
Statistical analyses
The statistical plan is explained in great detail.
- It is stated that Python statsmodels/pandas will be used for data analysis; articles or user manuals of these packages that underpin the methodology can be added as sources.
Discussion
This section summarizes the expected contributions, hypotheses, and possible limitations. However, the limitations section in this protocol is not clearly written. The article should include a short subheading such as "Limitations" or "Strengths and Limitations" and clearly state the above points. This increases the transparency of the protocol and protects against criticism when the results are published in the future:
- There is a single-center/multicenter ambiguity throughout the study. Initially, a single center (Hospital Central de la Defensa Gómez Ulla) was mentioned, followed by hospitals in Madrid. This limits generalizability. Furthermore, the sample size calculation has been made, but if the study were conducted at a single center, it would be difficult to achieve this number for a sample of 141 people at a single center.
- The method will involve numerous tests (psychological scales, EMG, blood samples, QST, etc.). This may lead to a high risk of dropout.
- The planned follow-up period is only 6 months. However, the long-term effects of surgical outcomes (1–2 years) may be significant.
Author Response
Dear Reviewer 2,
We sincerely appreciate your valuable comments and suggestions, which have significantly contributed to improving the clarity, methodological rigor, and clinical applicability of our manuscript. Below, we provide our responses, clearly separating each reviewer’s comment from our reply, while maintaining a fluent narrative style.
Comment 1: The study objective is not included in the abstract.
Response: Thank you for the observation. We have modified the abstract to explicitly add the main study objective at the end of the introduction. Lines 25-27
Comment 2: The biomarker section is disorganized, and the objective should be stated more concisely.
Response: Thank you for your comment. We have restructured the biomarker section to first present the most relevant markers (NfL, miR-30c-5p, clusterin), followed by their clinical rationale. In addition, we reformulated the objective into shorter and more direct sentences. Lines 90-127
Comment 3: Include information on the calculation of the FABQ (total score, interpretation high/low).
Response: Thank you for the suggestion. We have added in section 2.2.2 that the FABQ total score is obtained by summing the items of each subscale (physical activity and work), and that higher values indicate stronger avoidance beliefs, reflecting a greater risk of pain-related disability. Lines 182-199
Comment 4: The source of the effect size used in the sample size calculation is unclear. Why 0.35?
Response: We appreciate the reviewer’s observation. In the original submission we reported an effect size of f² = 0.35. Upon further consideration, and to adopt a more conservative and empirically grounded assumption, we have revised the calculation to use f² = 0.20 (R² ≈ 0.17). The manuscript has been updated accordingly. Lines 331-351
Comment 5: Excessive measurement tools-risk of dropouts. Is there a strategy?
Response: We thank the reviewer for this important observation. To minimize participant burden, all assessments will be clustered into a single comprehensive preoperative visit and one follow-up visit at 6 months. In addition, we will implement supportive strategies including telephone reminders, flexible scheduling, and assistance with transportation when needed. The research team will actively monitor adherence and dropout rates throughout the study, allowing us to adjust logistics promptly if unanticipated difficulties arise. We believe these measures will substantially reduce participant burden while ensuring the integrity and completeness of the data collected. Line 325-328
Comment 6: It is recommended to add references regarding the use of Python (statsmodels/pandas).
Response: We appreciate the reviewer’s suggestion. In the revised manuscript we have added references to the use of Python for statistical analysis. We chose to cite general references on Python rather than specific libraries (e.g., statsmodels or pandas), since library versions are not fixed and may evolve over time. This ensures methodological validity while maintaining the long-term relevance of the reference list. Line 332 and 352.
Comment 7: Limitations are not clearly written or subtitled.
Response: Thank you for your comment. We have added an explicit subsection titled “Limitations” within the Discussion, detailing: potential bias due to the high number of assessments, risk of dropout during follow-up, the 6-month follow-up period, and the challenge of reaching the target sample size within a single center. Line 422-441
Comment 8: Ambiguity about single-center/multicenter.
Response: Sorry for the misunderstanding. We have clarified that recruitment will take place at Hospital Central de la Defensa Gómez Ulla. Line 130-131
Comment 9: Limited follow-up of 6 months.
Response: Thank you for the comment. We acknowledge this limitation and have added it explicitly under the limitations section. We justify that, while studies with 1-2 years of follow-up are valuable, this initial protocol focuses on short-term outcomes (6 months). Future study phases may extend follow-up duration. Line 430-434
All reviewer comments have been incorporated, thereby enhancing the clarity, rigor, and clinical applicability of the manuscript. We once again thank the reviewers for their detailed and constructive feedback.
Sincerely,
Reviewer 3 Report
Comments and Suggestions for Authors
The manuscript addresses a clinically relevant and timely question: identifying prognostic determinants for outcomes in lumbar radiculopathy surgery, with a focus on biomarkers (NfL, microRNA-30c-5p, clusterin) alongside clinical and psychosocial factors. This integrative approach is valuable, as current models rely heavily on subjective measures.
- The novelty for lumbar radiculopathy is implied but not forcefully demonstrated.
- The manuscript does not sufficiently specify what is missing in current prognostic models and how this study uniquely fills that gap.
- Biomarker rationale appears in both the Introduction and Discussion, leading to redundancy and loss of focus.
- The abstract restates procedures instead of sharply highlighting research aims, hypotheses, and expected contributions.
- The effect size (0.35) used in sample size calculation is not well justified. Why moderate effect size? What prior data supports this?
- Inter-rater reliability or calibration procedures for EMG, algometry, and reflex testing are not addressed, though these are operator-dependent.
- Clusterin and microRNA-30c-5p are presented alongside NfL, but unlike NfL, these have very limited validation in lumbar radiculopathy.
- The review does not weigh contradictory findings or highlight uncertainty in biomarker research—only the promising side is emphasized.
- The extensive battery of tests (biomarkers, EMG, multiple questionnaires, QST, PPT, temporal summation, etc.) may be difficult to replicate in routine clinical settings.
Author Response
Dear Reviewer 3,
We sincerely appreciate your valuable comments and suggestions, which have significantly contributed to improving the clarity, methodological rigor, and clinical applicability of our manuscript. Below, we provide our responses, clearly separating each reviewer’s comment from our reply, while maintaining a fluent narrative style.
Comment 1: The novelty of lumbar radiculopathy is implicit but not strongly demonstrated.
Response: We thank the reviewer for this observation. We have expanded the introduction to emphasize the innovative nature of the study, noting that despite the high prevalence of lumbar radiculopathy, few studies integrate molecular biomarkers with clinical and psychosocial factors into a prognostic model. This integration represents an advance over previous models that rely almost exclusively on subjective clinical measures. Line 90-127
Comment 2: The manuscript does not sufficiently specify what is lacking in current prognostic models and how this study uniquely fills that gap.
Response: We thank the reviewer for this insightful comment. We have now expanded the Introduction to clearly specify the limitations of existing prognostic models—namely, their limited integration of biomarkers with psychosocial and clinical variables—and to highlight how our study uniquely addresses this gap by combining NfL, clusterin, and microRNA-30c-5p with validated psychosocial measures. We believe this addition clarifies the novel contribution of our protocol. Lines119-127
Comment 3: The justification for biomarkers appears in both the Introduction and Discussion, leading to redundancy and loss of focus.
Response: thank you for the observation. We have reduced redundancy. The Introduction now includes only the general rationale for why NfL, clusterin, and microRNA-30c-5p are relevant predictors. The Discussion maintains only the critical interpretation of our expected findings in relation to existing literature. Line 95-122
Comment 4: The abstract reaffirms procedures rather than clearly highlighting the research objectives, hypotheses, and expected contributions.
Response: We have rewritten the abstract to highlight: (1) the main objective (to identify prognostic determinants of lumbar radiculopathy surgery outcomes), (2) the hypothesis (that biomarkers and psychosocial factors will improve outcome prediction compared to clinical-only models), and (3) the expected contributions (development of an integrative model with potential clinical application). Line 23-48
Comment 5: The effect size (0.35) used in the sample size calculation is not well justified.
Response: We appreciate the reviewer’s observation. In the original submission we reported an effect size of f² = 0.35. Upon further consideration, and to adopt a more conservative and empirically grounded assumption, we have revised the calculation to use f² = 0.20 (R² ≈ 0.17). The manuscript has been updated accordingly. Lines 331-351
Comment 6: Reliability or calibration procedures between assessors for EMG, algometry, and reflex testing are not addressed.
Response: We thank the reviewer for this valuable observation. At present, no formal calibration procedures between assessors are planned for EMG, algometry, or reflex testing. All assessments will be performed by experienced clinicians following standardized protocols, which minimizes variability across raters. We acknowledge, however, that the absence of inter-rater calibration may represent a limitation of the study, and we have now explicitly stated this in the manuscript.
Comment 7: Clusterin and microRNA-30c-5p are presented alongside NfL, but unlike NfL, they have very limited validation in lumbar radiculopathy.
Response: We agree that evidence for clusterin and microRNA-30c-5p in lumbar radiculopathy is still limited. We have revised the text to present them as exploratory biomarkers. It is explicitly stated that the study aims to generate preliminary evidence and that their inclusion is based on their mechanistic potential in neuropathic pain rather than consolidated clinical validation. Line 435-441
Comment 8: The review does not weigh contradictory findings or highlight uncertainty in biomarker research; it only emphasizes the promising side.
Response: We have revised the Discussion to include contradictory findings and current limitations: variability in NfL reference values, heterogeneity of results in microRNA and clusterin studies, and the lack of replication in large samples. This underscores the need for studies like ours to validate or refute their clinical utility. Line 435-441
Comment 9: The extensive battery of tests may be difficult to replicate in routine clinical settings.
Response: We acknowledge this limitation and have made it explicit in the Discussion. We recognize that the broad assessment battery reflects the exploratory and research-oriented nature of the study. Nevertheless, we emphasize that the ultimate goal is to identify a reduced subset of predictors (biomarkers more key questionnaires) that can be translated into routine clinical practice. Line 415-434
All reviewer comments have been incorporated, thereby enhancing the clarity, rigor, and clinical applicability of the manuscript. We once again thank the reviewers for their detailed and constructive feedback.
Sincerely,
Round 2
Reviewer 1 Report
Comments and Suggestions for Authors
Most of the reviewer's suggestions and contributions have been addressed.
Reviewer 3 Report
Comments and Suggestions for Authors
The manuscript is suitable for publication in its current form